

# Zinc-reinforced ZSM-5 subject to a rare-earth magnet and the presence of a legume yields considerable copper extraction

Mouhammad Shadi Khudr[1], Cristian Baleca[2], Nasser Alqahtani[2,3], Hassan Alhassawi[2], Arthur Garforth[2], Gordon Tiddy[2,†], Abdullatif Alfutimie[2]

[1] Faculty of Biology, Medicine and Health, University of Manchester, Manchester, United Kingdom
[2] Chemical Engineering, University of Manchester, Manchester, Lancashire, United Kingdom
[3] Advanced Materials Technologies Institute, King Abdulaziz City For Science And Technology, Riyadh, Saudi Arabia
[†] Deceased author.

Corresponding author
Abdullatif Alfutimie,
abdullatif.alfutimie@manchester.ac.uk

## ABSTRACT

Heavy-metal extraction using physically modified zeolites with reducing agents, subject to static magnetism and bioremediation, remains largely unexplored. Here, we partly gel-coated ZSM-5 pellets with zinc and tested their copper extraction from a polluted medium, with and without a neodymium magnet and/or a bio-trap (*Vicia faba*). The reinforced zeolite accrued the fastest extraction, outperforming the raw zeolite and the bio-trap, especially as time advanced. The reinforced zeolite, accompanied by the bio-trap, was the most effective over time, followed by having both companions (the magnet and the bio-trap) present. Having the magnet as a solo companion of the reinforced zeolite extracted more copper compared to the raw zeolite; but slightly yielded lesser than having the reinforced companionless. Interestingly, the bio-trap was a better companion than the magnet, and, after 20 min, having both companions was more beneficial than having the magnet alone, but less yielding than having the bio-trap alone in the long term, indicating a synergistic effect between the reinforced zeolite and the bio-trap. Further characterisation underscored stable yet differential ZSM-5 performance. Additionally, we plasma-sputtered ZSM-5 pellets with zinc, and tested their copper extraction under the magnet. That led to promising early results, despite a sharply deteriorated extraction efficacy over time, as more extraction was achieved than the cases of the raw zeolite, or the reinforced zeolite minus the magnet, with almost identical outcome compared to the reinforced zeolite plus the magnet.
Our amalgamative approach provides novel user-friendly extraction methods, with high applicability potential across aquatic media and heavy metals.

## INTRODUCTION

### Compounded effects of copper pollution

A trace amount of copper is necessary for life to function, however, due to anthropogenic effects; $Cu^{2+}$ levels may exceed the permitted upper regulatory limit of 0.0015 g/L in drinking water and 0.0013 g/L in industrial effluents (*Bilal et al., 2013*). As copper has a high affinity for organic matter, it accumulates in sediments (*Han et al., 2001*; *Caillat et al., 2014*), so it does in biological systems (*Salt et al., 1995*; *Han et al., 2001*; *van Hullebusch et al., 2003*; *Chibuike & Obiora, 2014*; *Li et al., 2014*; *Saadani et al., 2016*; *Usman, Al-Ghouti & Abu-Dieyeh, 2019*). Exposure to excessive amounts of $Cu^{2+}$ can lead to various disruptive effects and impairment of biological processes (*Dave, 1984*; *Kramer et al., 2004*; *Lamb et al., 2012*; *Saadani et al., 2016*; *Igiri et al., 2018*; *Taylor et al., 2020*). Cupric stress can have inhibitory and detrimental effects on metabolism, growth, and development of plants in natural and agricultural systems (*Chibuike & Obiora, 2014*; *Shabbir et al., 2020*; *José Rodrigues Cruz et al., 2022*; *Schmitz et al., 2023*; *Xu et al., 2024*).

Sulfur is a non-metallic element vital to biogeochemical cycles and biosynthesis (*Giordano, Noriciand & Hell, 2005*; *Lucheta & Lambais, 2012*) and is amply available in different forms, such as sulfate $SO_4^{2-}$, which is prevalent in water (*World Health Organization (WHO), 2004*; *Giordano, Noriciand & Hell, 2005*; *Boone et al., 2012*). Sulfate levels in freshwater are generally around 20 mg/L (*World Health Organization (WHO), 2004*), whilst the Secondary Maximum Contaminant Level (SMCL) for sulfate in drinking water is 250 mg/L (*Webb & Czapar, 2013*). However, clear guideline values for sulfate in water have not been fully administered yet, see also (*Boone et al., 2012*). Excess amount of $SO_4^{2-}$ may concur with abundant $Cu^{2+}$ and thus aggravate the burdens of pollution on the environment due to various natural and anthropogenic processes (*Tabatabai, 1987*), such as the widespread use of copper sulfate $CuSO_4$ in fertilisers and pesticides as well as in water treatment (*van Hullebusch et al., 2003*; *Boone et al., 2012*).

Adsorption of anions, as well as heavy metal cations (*Vidal et al., 2009*; *Liu, Li & Zhang, 2010*; *Bilal et al., 2013*; *De Gisi et al., 2016*), represents an extraction technique that is affordable and generally environmentally friendly (*Montes-Atenas & Valenzuela, 2017*). Attaining improved remediation of pollution in ecosystems is, therefore, one of the main driving forces underlying the development of enhanced adsorbents (*Bilal et al., 2013*; *De Gisi et al., 2016*; *Virgen et al., 2018*).

### The employment of zeolites in the adsorption of pollutants

Zeolites, having the general formula ($M_{2/n}$ O·$Al_2O_3$·x$SiO_2$·y$H_2O$), n=cation valence (*Sherman, 1999*), are a group of natural or synthetic hydrated alumina-silicate minerals with reactive pores of molecular dimensions that allow configurational diffusion and thus entrapment of target anions or cations in the zeolite pores (*Masuda, 2003*; *Widayat & Annisa, 2017*). The zeolite polyhedral (tetrahedral) structure consists of a framework of $[SiO4]^{4-}$ and $[AlO4]^{5-}$ linked to each other at corners by sharing their oxygen atoms in a three-dimensional network with numerous voids (*Muraoka et al., 2019*). This crystalline microporous alumina-silicate structure varies according to the configuration of the oxygen

sharing (*Masuda, 2003*; *Muraoka et al., 2019*), where $Al^{3+}$ isomorphous substitution for $Si^{4+}$ in the said framework results in a negative charge that is key in the adsorption of heavy metal cations (*Erdem, Karapinar & Donat, 2004*; *Barakat, 2008*). The adsorption efficiency is subject to several factors including zeolite crystal structure, pore sizes, quantity, adsorbate concentration, pH, contact time, and affinity to desired adsorbates (*Margeta et al., 2013*).

ZSM-5 (Zeolites Socony Mobil-5) is a synthetic microporous zeolite of MFI framework type (*Priyadi & Mukti, 2015*; *Wojciechowska et al., 2019*), with a higher silica-to-alumina ratio, good thermal stability, and high contact surface area. With its unique framework structure, ZSM-5 possesses unique selectivity properties and thus is commonly used as an adsorbent and as a catalyst in the petrochemical industry (*Widayat & Annisa, 2017*; *Wojciechowska et al., 2019*; *Niu et al., 2022*; *Yuan et al., 2022*; *Zachariou et al., 2023*). These properties together with acid resistance and pores of 5.1–5.6 Å (*Shirazi, Jamshidi & Ghasemi, 2008*), make ZSM-5 a good adsorbent of heavy metals (*Wojciechowska et al., 2019*) and a suitable candidate for extracting $Cu^{2+}$ (1.28 Å diameter) from contaminated media.

As the porosity of ZSM-5 is key for its adsorption capacity (*Liu et al., 2021*), thermal, as well as chemical modification, are commonplace to increase mesoporosity and hence adsorption capacity (*Rac et al., 2020*). However, the adsorption ability of ZSM-5 has not received sufficient examination thus far (*Priyadi & Mukti, 2015*). The adsorption of heavy metals may improve by chemical modification of the zeolite *via* ion exchange or wet impregnation (*Kinger et al., 2000*; *Ciobanu et al., 2008*; *Wojciechowska et al., 2019*). The chemical modification can enhance the performance, selectivity, and functionality of the zeolite compared to the raw unmodified form (*Kinger et al., 2000*; *Cieśla et al., 2019*; *Wojciechowska et al., 2019*).

Chemically modified forms of ZSM-5 are used in desulfurisation through selective adsorption of sulfur containing compounds (*Garcia & Lercher, 1992*; *Liu, Li & Zhang, 2010*; *Liu et al., 2021*), *e.g.*, from refinery streams (*Dehghan & Anbia, 2017*; *Sarda et al., 2012*). Furthermore, the adsorption of pharmaceutical water contaminants has been achieved by *Rac et al. (2020)* using thermally and chemically treated de-silicated ZSM-5 with resultant increased mesoporosity. However, synthesis and chemical modification consume time and energy and are usually less environmentally friendly (*Muhammad, 2018*; *Cieśla et al., 2019*). Therefore, the production of physically modified zeolite forms through novel means with the potential to be more effective in capturing environmentally toxic pollutants is desirable. Interestingly, *Alswata et al. (2017)* demonstrated that hybridising zeolite with nano-particles of zinc oxide (ZnO), *via* co-precipitation, could yield considerable extraction of heavy metals from aqueous solutions. Nevertheless, physical rather than chemical modification of zeolites, such as ZSM-5, is still underdeveloped.

## Electrochemical series and magnetism
According to the electrochemical series, any metal with higher reactivity can be oxidised, whilst it reduces and displaces a lesser reactive metal from its soluble salt compounds.

For the displacement reaction to occur, the target species must be less reactive (*van Straten & Ehret, 1939*). For instance, by reacting with zinc solid $Zn_{(s)}$ (reducing agent), $Cu^{2+}$ of $CuSO_4$ (oxidising agent) accepts electrons donated by the $Zn_{(s)}$ and transitions accordingly from its aqueous state to a solid state, as shown in the following equations (*Sarda, Handa & Arora, 2016*):

$$Zn_{(s)} + Cu^{2+}_{(aq)} \rightarrow Zn^{2+}_{(aq)} + Cu_{(s)} \tag{1}$$

$$Zn_{(s)} \rightarrow Zn^{2+} + 2e^- \tag{2}$$

$$Cu^{2+} + 2e^- \rightarrow Cu_{(s)}. \tag{3}$$

It is striking, however, that modifying (*i.e.*, altering) the surface of zeolite using a reducing agent such as zinc (post-transitional metal) to accompany the sorption of heavy metal cations with the ability to electrochemically reduce the cations to metal solids has never been tested before despite being highly plausible. Such a dual functionality of pollutant extraction could be further enhanced, especially the electrochemical facet, by employing a static magnetic field. The latter has been demonstrated to accelerate and increase the displacement and deposition of copper on a zinc solid substrate (*Udagawa et al., 2014*). To date, an experimental test of capturing heavy metal cations using zeolites physically modified by a reducing metal, in a solid form, under strong magnetism represents a knowledge gap to fill.

## Phytoremediation

Phytoremediation is a common form of bioremediation (*Sen Gupta, Yadav & Tiwari, 2020*) that employs plants to extract/remove toxic pollutants (*Salt et al., 1995*; *Suman et al., 2018*; *Usman, Al-Ghouti & Abu-Dieyeh, 2019*). The main biological mechanisms of phytoremediation observed for heavy metal extraction are through uptake and deposition into biological mass in the plant root system (rhizofiltration) Furthermore, subsequent translocation of toxic metals to accumulate in the shoot system may occur, a process also referred to as phytoextraction (*Salt et al., 1995*). Tolerant plants are, therefore, useful in cleansing and phytostabilising affected surroundings by sequestering toxic cations (*Salt et al., 1995*; *Han et al., 2001*; *van Hullebusch et al., 2003*; *Chibuike & Obiora, 2014*; *Li et al., 2014*; *Saadani et al., 2016*; *Usman, Al-Ghouti & Abu-Dieyeh, 2019*).

Various wild or natural plants are shown to phytostabilise heavy metals (*e.g.*, *Xiao et al., 2008*; *Ahmadpour et al., 2015*; *Xu et al., 2019*; *Usman, Al-Ghouti & Abu-Dieyeh, 2019*), generally due to the negative charge of the cellular walls that can attract and compartmentalise heavy metal cations (*Jiang, Liu & Liu, 2001*; *Matijevic, Romic & Romic, 2014*; *Usman, Al-Ghouti & Abu-Dieyeh, 2019*). For example, when exposed to a group of metals including copper, a low shrub succulent, *Tetraena qataranse* (Hadidi) Beier & Thulin, accumulated 22.4 mg/kg of copper in the roots, whilst 6.2 mg/kg translocated into the shoots, compared to 5.7 mg/kg left in the soil (*Usman, Al-Ghouti & Abu-Dieyeh, 2019*). Moreover, plants are natural bio-adsorbents of sulfate (*Sadeghalvad et al., 2021*), as plants assimilate then store or metabolise the anion (*Calderwood & Kopriva, 2014*).

Interestingly, terrestrial plants can grow in aquaculture for the remediation of pollutants. For example, brown mustard, *Brassica juncea* (L.), and alpine pennycress, *Thlaspi caerulescens* (J.Presl & C.Presl), have been shown to reduce the levels of a combination of toxic metals, including copper, through phytoextraction, over several days of application (*Salt et al., 1995*). Whereas, through rhizofiltration *via* the plant root system, sunflower *Helianthus annuus* (L.) can substantially reduce copper levels amongst other heavy metals overnight (*Salt et al., 1995*). The Fabaceae family includes species that are generally tolerant accumulators of heavy metals (*Reeves et al., 2017*; *Yadav, Singh & Jadeja, 2021*). For example, the broad bean, *Vicia faba* (L.), is a popular global cash crop (*Karkanis et al., 2018*). This plant is rich in protein with robust shoots and roots where the root system colligates associated symbiotic nitrogen-fixing rhizobacteria, adding to the value of *V. faba* as green manure (*Hamdi, 1982*; *Chen et al., 2018*). Having considerable biomass, along with the said symbiotic association, corroborates the applicability of broad beans as a sustainable low-cost augmented phytoremediator of copper (*Teng et al., 2015*; *Saadani et al., 2016*). Moreover, *V. faba* has considerable tolerance to the uptake of metallic contaminants (*Abdel Hamed Abdel Latef & Abu Alhmad, 2013*; *Teng et al., 2015*), as the plant can phytostabilise and accumulate copper in its tissues (*Nadgórska-Socha et al., 2013*; *Matijevic, Romic & Romic, 2014*; *Alobaidi, 2016*), but that is relative to copper content in the medium (*Brennan & Bolland, 2003*). However, $Cu^{2+}$ concentrations of 0.159 g/L of $CuSO_4$ in hydroponic culture have been reported to have a genotoxic effect on *V. faba* after 42-h exposure (*Souguir et al., 2008*). Longer exposure to $Cu^{2+}$ from treatment with $CuSO_4$ results in oxidative stress, disruption of uptake of other elements, and inhibition of growth (*Mourato, Martins & Campos-Andrada, 2009*; *Abdel Hamed Abdel Latef & Abu Alhmad, 2013*; *Fatnassi et al., 2015*; *Alobaidi, 2016*; *Benouis et al., 2021*). That is in addition to necrosis that occurs at higher exposure levels ≥0.048 g/L (*Fatnassi et al., 2015*; *Alobaidi, 2016*). Trapping copper and sulfur using aquacultured *V. faba* combined with metal-reinforced zeolite subject to magnetism is yet to be examined.

This laboratory study presents a novel approach that amalgamates techniques from the fields of biology, physics, chemical engineering, and analytical science for pollutant extraction from freshwater contaminated with copper sulfate. By combining the use of zinc-reinforced ZSM-5 zeolite (through the physical embedding of zinc metal coarse powder on part of the zeolite surface), a companion rare-earth (neodymium) static super magnet, and/or an accompanying aquacultured faba bean, we provide a timely and novel solution to treat heavily polluted aquatic systems. See Fig. 1 for the experimental setup and design. The current article is peer-reviewed based on our revised preprint (*Khudr et al., 2023*).

## MATERIALS AND METHODS

### Experimental set-up

The aqueous medium of the experiment mimicked freshwater used in aquaculture (*Klüttgen et al., 1994*). The medium was standardised, reproduced in bulk, stored in the lab environment, and aerated with an aquarium air pump for 12 h before the experiment. The medium consisted of a mixture of 0.33 g/L of sea salt (Sigma Aldrich©, Gillingham,

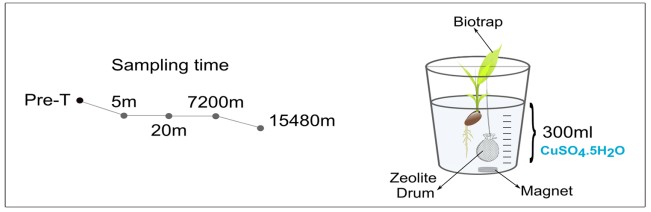

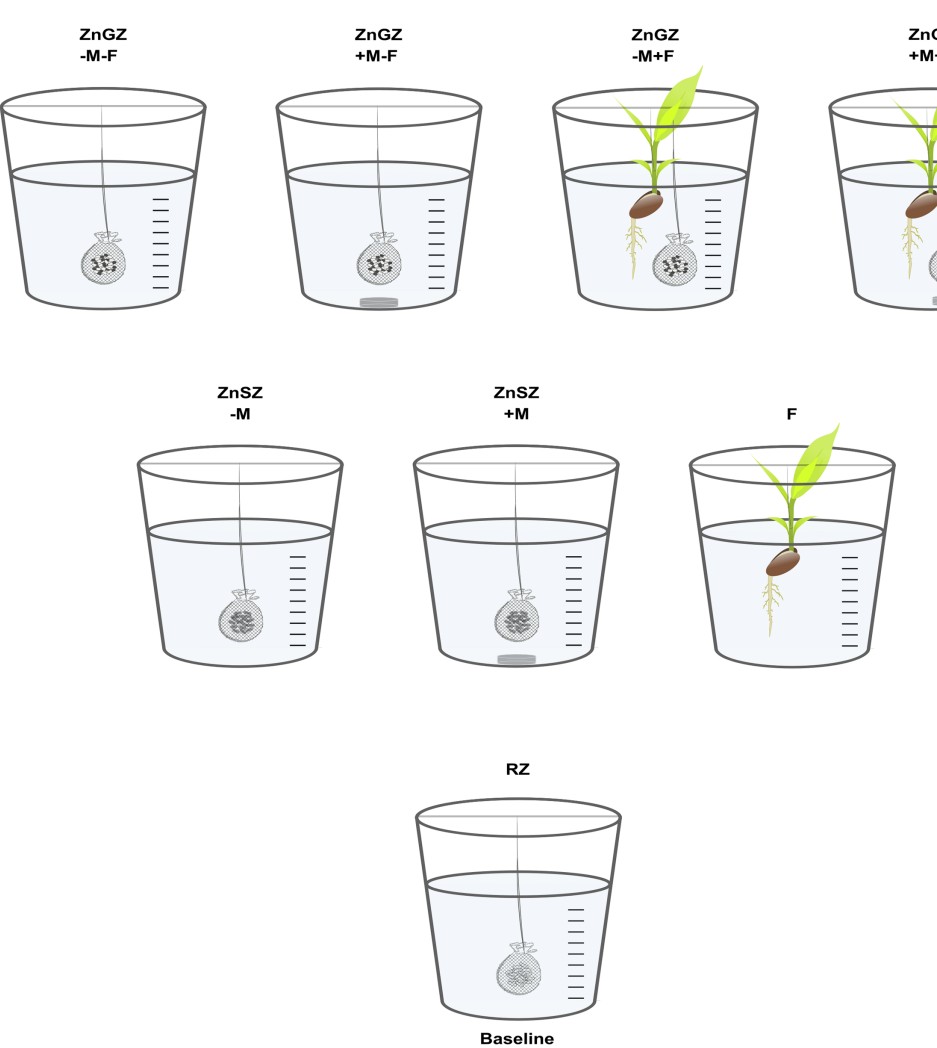

**Figure 1 Experimental setup.** The set-up of nine combinations that were tested respectively against copper extraction over time. The initial copper concentration was 0.178 g/L in the target medium polluted with copper sulfate pentahydrate $CuSO_4.5H_2O$. There were eight treatment combinations: raw zeolite (RZ) as the baseline, the bio-trap (*V. faba*) alone (F), zinc-gel-reinforced zeolite minus magnet without *V. faba* (ZnGZ-M-F), zinc-gel-reinforced zeolite plus magnet without *V. faba* (ZnGZ+M-F), zinc-gel-reinforced zeolite plus magnet with *V. faba* (ZnGZ-M+F), zinc-gel-reinforced zeolite plus magnet with *V. faba* (ZnGZ+M+F), zinc-sputtered zeolite minus the magnet (ZnSZ-M), and zinc-sputtered zeolite plus the magnet (ZnSZ+M). The magnet used was made of neodymium. There were three repeats per treatment and four sampling times in minutes (5, 20, 7,200 and 15,840). Pre-T = time zero prior to the treatment.

UK), 23 ml/L of 117.6 g/L calcium chloride dehydrate $CaCl_2.2H_2O$ (Fisher Scientific©, Loughborough, UK), 22 ml/L of 25.2 g/L sodium bicarbonate $NaHCO_3$ (Honeywell©, Strahlenbergerstr, Germany), and 1 ml/L of 0.07 g/L selenium dioxide $SeO_2$ (Merck KGaA©, Darmstadt, Germany) dissolved in Millipore Milli-Q de-ionised high purity water (pH = 7).

The pollutant was dry crystal copper sulfate pentahydrate $CuSO_4 \cdot 5H_2O$ (Sigma Aldrich©, Gillingham, UK). Imitating circumstances of potential leakage or spillage associable with anthropogenic activities, 0.7 g/L of the pollutant was applied. Copper extraction was conducted using a complex treatment: (1) zinc-reinforcement of zeolite: The zeolite of choice (ZSM-5) was applied in its raw (zinc-free form) as the baseline, or zinc-reinforced forms (*via* partial zinc gel coating or zinc plasma sputtering), −/+ rare-earth magnet, in the absence of aquacultured faba bean plant *V. faba*. (2) *Vicia faba* as the model bio-trap of $Cu^{2+}$ was applied alone. (3) The zinc-gel-reinforced form (−/+ the magnet) was accompanied by the *V. faba* bio-trap (Fig. 1).

## Zeolite reinforcement

Commercial grade raw zeolite ZSM-5 pellets, with Si/Al of 50, were acquired by the Department of Chemical Engineering (CE), The University of Manchester, from Pingxiang Naike Chemical Industry Equipment Ltd© (Pingxiang, Jiangxi, China). These zeolite pellets are stable synthetics. An amount of 7 g of bacterial agar powder (Sigma Aldrich©, St. Louis, MO, USA) was used to reinforce half of the surface of the zeolite pellet with coarse zinc powder, particle size about 0.3–1.5 mm (14–50 mesh ASTM), EMSURE® Reag. Ph Eur (Sigma Aldrich©, Gillingham, UK). The gel powder was dissolved in 100 ml of high-purity deionised water then heated on a hot plate and stirred continuously until it became homogeneous and viscous. Subsequently, ZSM-5 pellets were systematically semi-immersed individually in the done gel, using a forceps, and then coated/dusted with 0.5 g of zinc coarse powder. The resulting zinc-gel-reinforced zeolite pellets were left to dry for 4 days to allow the gel to solidify into a hard resin-like structure, holding the zinc fine metal pieces firmly onto the gel-surfaced part of the zeolite. The experiment took place at room temperature and the pH and conductivity were measured several times during the experiment using a Eutech PC 2700 probe (Thermo Fisher Scientific, Waltham, MA, USA).

Additionally, to reinforce the zeolite with nano-layers of zinc (Testbourne©, Whitchurch, UK), raw ZSM-5 was plasma-sputtered with zinc longitudinally on half of the zeolite pellet surface using a coating machine based on the technology of scanning electron microscopy (FEI Quanta 200 Environmental, FEI Company©, Hillsboro, OR, USA). The sputtering technique allows using an argon plasma stream under a vacuum to separate zinc metal atoms off a zinc disc target by applying a high voltage to initiate plasma gas ion exchange. The plasma sputter anchored the active metal (zinc) atoms onto the surface of the ZSM-5 zeolite pellets as an ultra-thin film. Technical difficulties in materialising this approach using the said machine and the substrate zinc disc resulted in having a very limited number of repeats for this addendum and hence the outcome of the zinc sputtering of zeolite is anecdotal. The zinc-sputtered zeolite treatments (ZnSZ-M) and (ZnSZ+M) were, therefore, excluded from the statistical analysis specified below.

One gram of the zeolite ZSM-5 (raw or zinc-reinforced) was wrapped in a customisable sachet of ultra-fine polyester mesh (~3-micron pores), acquired from JoTech Ltd©, Warwickshire (UK). The mesh enveloping the zeolite was used to create what is termed herein as the 'drum' (extraction device), and was employed in every treatment involving either of the zeolite forms. The drum was tethered to a scaffold that enabled it to be submerged in 300 ml of the experimental medium per beaker. Beakers were cling-filmed to control evaporation and 4-ml pipettes were respectively used to penetrate the respective beakers through the resealable cling film, where the medium was pipette-pumped multiple times every sampling time (see below) to homogenise the solution and mimic semi-stagnant water conditions; then the beakers were resealed accordingly.

## Preparation of the model bio-trap of copper

Seeds of one cultivar of faba bean *V. faba var minor* (Harz) (local supplier, Manchester, UK) were used to grow *V. faba* plants as a model organism for the extraction of $Cu^{2+}$ from the experimental medium. *Vicia faba* in aquaculture was chosen as a innovative means to extract and immobilise $Cu^{2+}$, building on the fact that this legume is rich with proteins to which copper cations may have a great affinity, and the plant is easy to grow and maintain with robust shoots and roots.

The plants were grown individually in 9-cm round plastic pots using soil compost (Levington F2) and watered when needed. Around 10 days post-germination, each plant was carefully extracted from the compost, the root system was washed thoroughly in water baths to remove compost debris, and then each plant was transferred to a beaker containing 300 ml of the experimental medium. Our pilot trials showed that the plant adapted to the contaminant-free aquatic medium with no signs of root degradation and exhibited significant continuous growth over 14 days. The top of each beaker, where the plant root was submerged, was cling-filmed to minimise evaporation; the plant stem emerged through the cling film. Each plant was kept upright through suspension using a thread attached to a metallic scaffold, with the plant root fully submerged in the experimental medium (aquaculture) (Fig. S1). The plants were supplied with organic spirulina powder (Naturya©, Bath, UK) dissolved and homogenised in the control experimental medium The latter supplement 2 ml of (1 g/L) was injected into the medium every three days through the beaker cling-film seal.

## Application of rare-earth magnet

A neodymium magnet disc (25 mm × 10 mm × 6 mm) with a vertical pull of 14.8 kg (N42, ~1.2 Tesla) (First4Magnets®, Tuxford, UK) was employed. Neodymium magnets, permanent potent static magnets made from rare-earth element alloys, are quite affordable and suitable for use at room temperature. The magnet was cling-filmed, to avoid corrosion, and then placed it in the centre of the beaker base underneath the submerged zeolite drum for 180 min starting from the beginning of the experiment. This was done to stimulate copper deposition on the zinc-reinforced surfaces of the zeolite, drawing on the rapid deposition process shown by Udagawa et al. (2014), see Fig. 1 for an illustration of the design. After the time-bound application, the magnet was carefully removed without

affecting the integrity of the zeolite drum. The zinc-reinforced zeolite forms (*via* gel adhesion or plasma sputtering) were tested with and without the presence of the magnet. For the treatment levels where the *V. faba* bio-trap accompanied the zinc-gel-reinforced zeolite, the root of the hanging plant was placed in close proximity to the magnet, while the zeolite drum was submerged directly over the magnet (Fig. S1). The zeolites (raw or zinc-gel-reinforced) were tested concurrently with the application of the aquacultured *V. faba*, with and without magnet presence. Due to the aforementioned limitations with the sputtering, the application of the zinc-sputtered zeolite was not tested in the presence of *V. faba*.

## Characterisation and analysis of copper extraction and materials

Energy Dispersive X-ray Spectrometer (EDAX) was used to perform a chemical microanalysis of the zeolite elemental composition and to assess $Cu^{2+}$ extraction and accumulation onto the zeolite. We characterised the raw unmodified form of the ZSM-5, the *zinc-free* bare part of the reinforced zeolite, and the *zinc-treated* (*zinc-reinforced*) surface of the partly reinforced zeolite. The comparative characterisation helped measure copper extraction and whether the presence of zinc supported or inhibited copper adsorption by the zinc-free part of the zeolite. This was also accompanied by using a Scanning Electron Microscopy-Energy Dispersive X-ray Spectrometer (SEM/EDX) (FEI Quanta 200 Environmental; FEI Company©, Hillsboro, OR, USA) to visualise the morphology/structure of ZSM-5 and to further inspect areas in the zeolite surface for element accumulation and dispersion (copper in particular). Furthermore, supportive characterisation of the zeolite (before and after application) and across different treatments was carried out. X-ray Powder Diffraction (XRD) was done to examine the crystal structure of ZSM-5, using a X-ray diffractometer D8 Discover (Bruker©, Coventry, UK) at room temperature using Cu Kα radiation (λ = 1.5406 Å), from 5 to 90 2θ in 0.02° steps, with 1 s per step with 30 mA and 40 kV generator. Additionally, BET analysis was done to measure the surface area and pore volume of the zeolite using a Micromeritics ChemiSorb 2750 analyser (Micromeritics©, Norcross, GA, USA) at −196 °C, using an argon–helium mixture (10 vol % argon). ZSM-5 samples (~150 mg) were degassed at 300 °C under a vacuum (500 µm) for 240 m. The zeolite samples were preliminarily held in helium at 200 °C for 1 h.

An Inductively Coupled Plasma Optical Emission Spectrometry (ICP-OES) (Vista-PRO, Varian Inc©, Palo Alto, CA, USA) was used to measure the residual concentration of $Cu^{2+}$, relative to the initial copper concentration (0.178 g/L), per treatment over time. A regression test (generalised linear model, glm) in the R environment with a quasiPoisson family (*R Core Team, 2021*), using the package 'multcomp' was applied (*Hothorn, Bretz & Westfall, 2008*). The explanatory variables were: (1) Treatment effect (six levels: raw zeolite (RZ) as the baseline, the bio-trap (*V. faba*) alone (F), zinc-gel-reinforced zeolite minus magnet without *V. faba* (ZnGZ-M-F), zinc-gel-reinforced zeolite plus magnet without *V. faba* (ZnGZ+M-F), zinc-gel-reinforced zeolite plus magnet with *V. faba* (ZnGZ-M+F), zinc-gel-reinforced zeolite plus magnet with *V. faba* (ZnGZ+M+F), (2) sampling time (four levels in minutes: 5, 20, 7,200 and 15,840), (3) the interaction (Treatment X sampling

time). The total number of repeats across the treatments analysed statistically was 69, as a limited number of repeats was discarded due to machine or personal errors and/or necrosis of the exposed faba bean bio-trap. The sampling times were adopted based on: (1) Influenced by a neodymium magnet, copper displacement by zinc occurs rapidly (*Udagawa et al., 2014*) then plateaus quickly, justifying the 5 and 20 min samples. (2) Copper uptake by the bio-trap is gradual and accumulative, justifying later samples at 7,200 and 15,840 min. (3) To avoid detrimental effects of high copper exposure on the bio-trap, the 15,840 min sampling was applied based on our pilot. (4) The treatment lines of the current multifactorial experiment were run simultaneously with systematic sampling.

## RESULTS AND DISCUSSION

The SEM-EDS microanalysis revealed a differential deposition of copper on the zinc-reinforced as well as the bare part of the zeolite pellets. The top three highest amounts of post-treatment copper deposition were detected as follows: 17.1% weight pertaining to the pellet core as per the treatment (ZnGZ-M-F), followed by 7.65% weight pertaining to the zinc-coated area of the pellet of the treatment (ZnGZ-M-F), then 5.1% of the zinc-coated area of the pellet of the treatment (ZnGZ+M-F). Whereas, the least detected amount (0.24% weight) was recorded in the uncoated area of the pellet of the treatment (ZnGZ+M-F) (Table 1). This supports the findings reported by *Priyadi & Mukti (2015)*, describing a positive correlation between increasing metal ion concentration and increased mobility of the surface-adsorbed metal inwards into the zeolite pores that act as molecular sieves (*Flanigen, 2001*; *Song et al., 2015*). The adsorption as well as filtering ability of the applied zeolite led to decreased concentration of the target ions, which is in line with the findings reported by *Priyadi & Mukti (2015)*, as corroborated by the present ICP-OES findings illustrated below.

More copper deposited on the zinc-reinforced surface of the zeolite when compared to the deposition on the zinc-free part or the raw zeolite; the deposition was, however, higher in the case of the zinc-reinforced zeolite when the magnet was absent (Table 1). Interestingly, the element microanalysis also indicated that the combination (ZnGZ-M-F) led to a notable extraction of sulfur (3.93% weight) in the presence of the magnet, as detected in the pellet core. Note that the reading for the zinc-reinforced area of the same combination was (0.89% weight), a stark contrast with the case when the magnet was present in (ZnGZ+M-F). However, the value of sulfur for the zinc-free pellet surface was (0.37% weight) (Table 1). The dynamics of sulfur adsorption on the bare zeolite surface and copper displacement and deposition (*Udagawa et al., 2014*), on the zinc-coated surface was clearly influenced by the magnet presence (Table 1). Nevertheless, the findings here are in line with those reported by *Liu et al. (2021)* on the good potential of ZSM-5 applicability to adsorb sulfuric compounds and treat polluted water. See Figs. S2–S5 for SEM images of the pellets pre- and post-treatment.

The XRD analysis characterisation before and after application confirmed stability in the crystal structure of ZSM-5 across treatments, as illustrated in Fig. S6, where 2θ peaks were aligned, with no changes in peak breadth, indicating a match between the XRD patterns. This was underscored by the modification/augmentation of the zeolite surface

**Table 1  EDAX microanalysis of copper and sulfur.**

| Copper and sulfur capturing material | Element | Cu | S |
|---|---|---|---|
| Raw zeolite (RZ-F) | Weight% | 1.09 | 0.37 |
| Zinc-free area of zinc-gel-reinforced minus magnet (ZnGZ-M-F) | Weight% | 2.48 | 0 |
| Zinc-reinforced area of zinc-gel-reinforced minus magnet (ZnGZ-M-F) | Weight% | 7.65 | 0.89 |
| Pellet core of the zinc-gel-reinforced minus magnet (ZnGZ-M-F) | Weight% | 17.10 | 3.93 |
| Zinc-free area of zinc-gel-reinforced plus magnet (ZnGZ+M-F) | Weight% | 0.24 | 0.37 |
| Zinc-reinforced area of zinc-gel-reinforced plus magnet (ZnGZ+M-F) | Weight% | 5.10 | 0 |

**Note:**

Results of the EDAX analysis are shown for each treatment. RZ-F refers to raw zeolite without *V. faba*, F refers to *V. faba* alone, and ZnGZ-M-F refers to zinc-gel-reinforced zeolite minus magnet without *V. faba*, ZnGZ+M-F refers to zinc-gel-reinforced zeolite plus magnet without *V. faba*; Number of iterations = 6. All pre-treatment readings for Cu and S were nil.

which was physical and partial. All patterns illustrated are in line with those reported by *Shirazi, Jamshidi & Ghasemi (2008)*. See Fig. S6 for further information.

Compared to the status before application/treatment, the BET analysis indicated a noticeable increase in the surface area ($m^2$/g) of the zinc-free part of ZSM-5 in all cases when the zeolite surface was partly reinforced with zinc, except for the case of the raw zeolite (non-augmented). This could be attributed to the increased extraction of copper through displacement and deposition on the zinc-coated part of the zeolite compared to the zinc-free part *i.e.*, the functionally adsorbing part, as confirmed by the microanalysis shown in Table 1. A decreasing number of cations was available to adsorb on the zinc-free part per se and thus bigger pore volume and more surface area were detected. Remarkably, all treatments with zinc-reinforced ZSM-5 led to significant extraction of copper from the target medium over time as shown by the ICP-OES analysis below. In other words, the partial zinc coating (reinforcement) of the zeolite pellet, created a new active surface with a different and also superior copper extraction capacity than the adsorptive bare one.

Less surface area and pore volume were detected in the presence of the magnet as a sole companion of the zinc-reinforced zeolite when compared to the case when the magnet was absent. Conversely, the presence of the bio-trap companion was always associated with more surface area and pore volume, indicating a synergistic effect of the *V. faba* bio-trap in filtering/extracting the ionic pollutants that facilitated less pore occupation on the interactive surfaces of the zeolite. As such, the highest increase was detected in the combination (ZnGZ+M+F), while the lowest was detected in (ZnGZ+M-F), yet still higher than the value when the raw zeolite was applied alone; a similar trend was seen regarding the pore volume (Table 2). Since ZSM-5 has larger silica to alumina content, the crystallinity of ZSM-5 is inherently declined, with smaller specific surface areas and pore volume (*Yuan et al., 2022*), influencing its sorption capacity (*Shirazi, Jamshidi & Ghasemi, 2008*). This usually necessitates thermal and chemical treatments to increase porosity and enhance adsorption (*Rac et al., 2020*). As such, the current work presented here is a promising alternative in copper extraction, whilst it spares the need for the conventional thermochemical route.

**Table 2 BET analysis of ZSM-5.**

| Sample | Micropore area (m²/g) | External surface area (m²/g) | Total surface area (m²/g) | Pore volume (cm³/g) | Pore size (Å) |
|---|---|---|---|---|---|
| RZ before treatment | 145.3997 | 104.769 | 250.1687 | 0.076366 | 23.39 |
| RZ after treatment | 146.9395 | 81.1129 | 228.0525 | 0.07721 | 24.377 |
| ZnGZ-M-F | 173.9235 | 102.6923 | 276.6157 | 0.091219 | 23.359 |
| ZnGZ+M-F | 164.7671 | 92.1266 | 256.8937 | 0.086508 | 22.498 |
| ZnGZ-M+F | 192.7065 | 100.0884 | 292.7949 | 0.101206 | 20.787 |
| ZnGZ+M+F | 202.4255 | 135.5142 | 337.9397 | 0.106289 | 23.643 |

**Note:**

Measurements of BET surface area, pore volume, and pore size are shown for the zeolite ZSM-5 across treatments. The target medium was polluted with copper sulphate pentahydrate $CuSO_4.5H_2O$ (0.7 g/L). Raw zeolite (RZ) before treatment (baseline), zinc-gel-reinforced zeolite minus neodymium magnet without the bio-trap *V. faba* (ZnGZ-M-F), zinc-gel-reinforced zeolite plus the magnet without *V. faba* (ZnGZ+M-F), zinc-gel-reinforced zeolite plus the magnet with *V. faba* (ZnGZ-M+F), and zinc-gel-reinforced zeolite plus the magnet with *V. faba* (ZnGZ+M+F).

The ICP-OES analysis revealed that in the absence of the magnet and the *V. faba* bio-trap, the treatment with zinc-gel-reinforced zeolite (ZnGZ-M-F) yielded the fastest as well as most effective extraction of $Cu^{2+}$, as ~78% of $Cu^{2+}$ (referred to here as residual) was left in the medium by 5 min. Then, at 20 min, the said treatment also led to the best outcome across treatments (~77% residual $Cu^{2+}$) (Fig. 2; Table S1). Overall, the produced partially zinc-augmented ZSM-5 effectively mitigated pollution *via* two simultaneous mechanisms of copper extraction. The first is the adsorption of $Cu^{2+}$ due to the permeability and active sites of the zinc-free mesoporous surface (*Visa & Popa, 2015*; *Priyadi & Mukti, 2015*; *Renu & Singh, 2017*; *Hussain, Madan & Madan, 2021*). Whereas, the second is the displacement and deposition of copper on the zinc coating of the augmented part of the zeolite surface, with and without the magnet effect (*van Straten & Ehret, 1939*; *Udagawa et al., 2014*; *Sarda, Handa & Arora, 2016*).

Pollutant removal from water by ZSM-5 is contingent on pH, pore structure, and hydrophobicity (*Shirazi, Jamshidi & Ghasemi, 2008*; *Liu et al., 2021*; *Yuan et al., 2022*). Since there is more silica to aluminium in ZSM-5, these factors underlaid the inferior performance of the raw ZSM-5, when compared to the zinc-reinforced, as they affected the interaction between ZSM-5 and water, including the adsorbates (*Jentys et al., 1989*; *Liu et al., 2021*; *Yuan et al., 2022*). This provides some explanation of the factors that influenced the differential outcome of copper extraction across treatments over time. Further, the partial reinforcement of the zeolite with zinc in our method counterbalanced the decrease in adsorption sites and strength of interaction with water (*Jentys et al., 1989*). As it did in offering a rewarding alternative for chemical treatment conventionally applied to increase selectivity (*Jentys et al., 1989*) and mesopore formation (*Rac et al., 2020*).

However, at 7,200 min, copper extraction peaked where the faba bio-trap accompanied the zinc-gel-reinforced zeolite in (ZnGZ-M+F), (only ~24% residual $Cu^{2+}$), as the treatments (ZnGZ-M-F) and (ZnGZ+M+F) were not far behind with (~33% residual $Cu^{2+}$) and (~34% residual $Cu^{2+}$), respectively. At 15,480 min, the best reading was under (ZnGZ-M+F), (~16% residual $Cu^{2+}$), which was followed by (ZnGZ+M+F), (~22% residual $Cu^{2+}$) (Fig. 2; Tables S1, S2). These outcomes, where the plant was present, are attributable to a composite action of copper extraction through the discussed inorganic

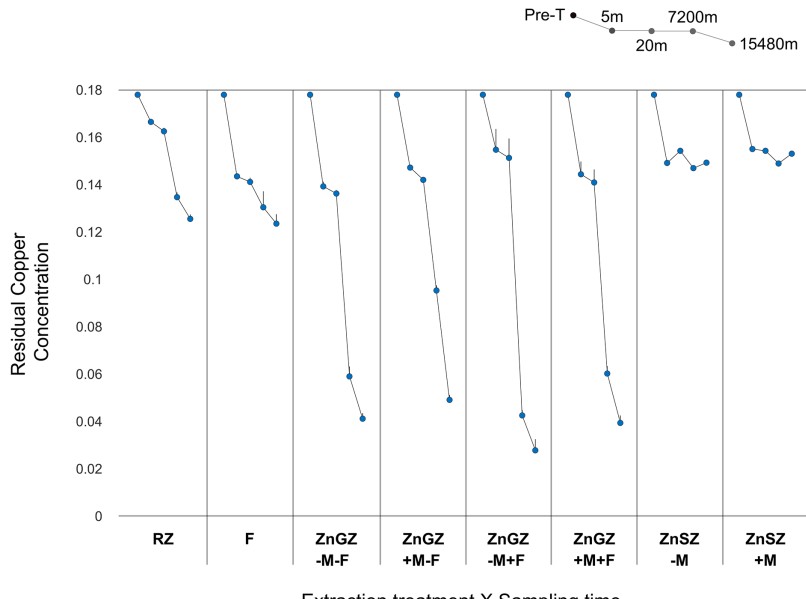

**Figure 2 Copper extraction.** Results of residual $Cu^{2+}$ concentration (mean ± SEM) in the treated medium per treatment lines are shown. The initial copper concentration was 0.178 g/L in the target medium polluted with copper sulfate $CuSO_4.5H_2O$ (0.7 g/L). The treatment combinations were raw zeolite alone (RZ), the bio-trap (*V. faba*) alone (F), zinc-gel-reinforced zeolite minus magnet without *V. faba* (ZnGZ-M-F), zinc-gel-reinforced zeolite plus magnet without *V. faba* (ZnGZ+M-F), zinc-gel-reinforced zeolite plus magnet with *V. faba* (ZnGZ-M+F), and zinc-gel-reinforced zeolite plus magnet with *V. faba* (ZnGZ+M+F). There were three repeats per treatment and four sampling times in minutes (5, 20, 7,200 and 15,840), Pre-T = time zero prior to the treatment.

method along with phytoremediation by the bio-trap (*Salt et al., 1995*; *Nadgórska-Socha et al., 2013*; *Matijevic, Romic & Romic, 2014*; *Alobaidi, 2016*; *Ghazaryan et al., 2019*; *Usman, Al-Ghouti & Abu-Dieyeh, 2019*; *Sen Gupta, Yadav & Tiwari, 2020*).

The zinc-gel-reinforced zeolite (ZnGZ-M-F) dramatically outperformed the raw zeolite (RZ) and the *V. faba* bio-trap (F) at every sampling time, particularly at later stages of the experiment (Fig. 2; Table S1). Similarly, the presence of the magnet alone in (ZnGZ+M-F) surpassed in its outcomes those of the raw zeolite at every sampling time, yet always with lesser copper extraction when compared to its absence (ZnGZ-M-F) (Fig. 2; Tables S2, S3). The results here corroborate those reported by (*Virgen et al., 2018*) on the magnet effect leading to a notable improvement in metal reactivity and facilitated accumulation and deposition of heavy metal ions onto the zeolite surface. In addition, our findings also lend support to those documented by *Rajczykowski & Loska (2018)* regarding improved copper extraction, exceeding 40% within 1 h, using activated carbon under the influence of a strong static magnetic field (N38, 0.517 T). Nevertheless, the underlying mechanism behind this phenomenon still requires further research to demystify.

The zeolite did not show signs of degradation from either the physical modification with zinc or contact with the copper sulfate solution. This is supported by the outcomes of the XRD tests, showing crystalline structure stability across treatments despite minor

detectable differences in peak intensities (Fig. S6). In other words, the metal loading did not negatively affect the unmodified area of ZSM-5. As such, a trade-off is clear, since the partial gelled zinc reinforcement of the zeolite surface reduced access to zeolite reactive pores on the reinforced part and hence minimised the adsorption capacity of the said surface. Nevertheless, the zinc reactivity on the reinforced surface counterbalanced that and paid off by forcing extra cationic copper to electronically convert to $Cu_{(s)}$ and deposit on the zinc substrate.

Interestingly, having the bio-trap alone in (F) led universally to better outcomes than those of the raw zeolite (Fig. 2; Table S1). That was more apparent at 5 and 20 min, as the copper would show high affinity to the organic materials of the plant tissues whereby it accumulates over time (*Nadgórska-Socha et al., 2013*; *Matijevic, Romic & Romic, 2014*; *Alobaidi, 2016*). The plant acts as porous ion-exchangers akin to gels and porous oxides, whilst it immobilises heavy metal cations (*Allan & Jarrell, 1989*) such that $Cu^{2+}$ accumulates gradually in the plant system (*Nadgórska-Socha et al., 2013*; *Matijevic, Romic & Romic, 2014*; *Teng et al., 2015*; *Alobaidi, 2016*; *Saadani et al., 2016*). Approaching the end of the experiment, constant exposure to the polluted medium led to a negative impact on the bio-trap, since the plants are prone to gradually developing necrosis throughout exposure (*Souguir et al., 2008*; *Mourato, Martins & Campos-Andrada, 2009*; *Abdel Hamed Abdel Latef & Abu Alhmad, 2013*; *Fatnassi et al., 2015*; *Alobaidi, 2016*; *Benouis et al., 2021*). The bio-trap in the current study was severely exposed to copper pollution for a relatively prolonged period of 11 days; the exposure started at 0.178 g/L on Day 1 and then reached a mean of ~0.124 on Day 11 when the bio-trap was alone in the experimental medium. For a relative measure, the copper extraction by the bio-trap in (F) on Day 11 was ~199% less effective than that of the zinc-reinforced zeolite in (ZnGZ-M-F) and ~342% less effective than that of the combination of the zinc-reinforced zeolite and the bio-trap (ZnGZ-M+F) (Tables S2, S3). That is also still dramatically higher than the level of ≥0.048 g/L associated with the induction of necrosis (*Abdel Hamed Abdel Latef & Abu Alhmad, 2013*; *Fatnassi et al., 2015*; *Alobaidi, 2016*). This could explain the lesser ability to trap copper alone over time by the bio-trap if not supported by an inorganic method, *i.e.*, zinc-reinforced zeolite in our example, as highlighted below. No signs of necrosis were seen when the *V. faba* bio-trap was coupled with the zinc-reinforced zeolite (with and without magnet presence), see Supplemental Information including Figs. S7–S9 for supportive details on plant parameters and biochemical responses to cupric stress using Fourier transform infrared spectroscopy (FT-IR) following freeze-drying (lyophilisation).

Copper extraction by the bio-trap in (F) at later stages of the experiment was close to the raw zeolite (RZ), but (F) was inferior to the zinc-augmented zeolite (ZnGZ-M-F) at every sampling time; remarkable inferiorities were seen in the long run. Nevertheless, the addition of the bio-trap to the reinforced zeolite without the magnet (ZnGZ-M+F) led to the best outcomes of (~24% residual $Cu^{2+}$) at 7,200 min and (~16% residual $Cu^{2+}$) at 15,480 min, but not below these sampling times (thresholds). As such, in the long term, accompanying the zinc-reinforced zeolite with the bio-trap is sizeably more yielding of copper extraction than having the magnet as a sole companion (Fig. 2; Tables S2, S3).

Further, having both the magnet and the bio-trap to accompany the reinforced zeolite was universally more yielding, beyond 20 min, than the raw zeolite, and the combined effects of the reinforced zeolite and the bio-trap were bigger than their individual effects after 20 min. When compared to the reinforced zeolite alone (ZnGZ-M-F), the presence of both the magnet and the bio-trap led to better outcomes at 5 and 20 mins than having either of them as a sole companion. However, as time progressed, that was only applicable comparative to the magnet case (ZnGZ+M-F), since the positive effect of the *V. faba* bio-trap as a single companion in (ZnGZ-M+F) manifested only after 20 mins, exceeding those of the other treatments (Fig. 2; Tables S2, S3).

In other words, the magnet effects were short-lived due to an unknown mechanism, but possibly either the restricted 3 h application, as described in the Methods or the retarded effect of the bio-trap overtime, or both played a role (Tables S2–S4). The aforementioned interpretations of the magnet effect, the trade-off, and the dual inorganic-organic action provide further explanation for the higher copper-extraction efficiency of the treatments (ZnGZ-M+F) and (ZnGZ+M+F). For reference, *Priyadi & Mukti (2015)* reported a considerable copper adsorption capacity (69.93 mg/g over 250 mins) of a chemically enhanced ZSM-5 (ratio Si:Al of 100, saturated with 0.5M of NaCl for 24 h), influenced by the zeolite pore size, negative charge, and the diameter of $Cu^{2+}$ (*Priyadi & Mukti, 2015*). Comparatively, our results show a highly competitive copper-extraction capacity of the physically modified ZSM-5, with and without the companionship with the *V. faba* bio trap.

Lastly, the zinc sputtering of the zeolite showed similar results in the absence and presence of the magnet, (ZnSZ-M) and (ZnSZ+M), respectively. The sputtering led to considerably better extraction of copper, than (RZ), at 5 min, and that also exceeded the reading of (ZnGZ-M+F) at that sampling time in the absence of the magnet (ZnSZ-M), or was on par with (ZnGZ-M+F) when the magnet was present (ZnSZ+M) (Fig. 2). The sputtering readings, with or without magnet, at 20 min, were also close to those of (ZnGZ-M+F). Nevertheless, the performance of the sputtered zeolite deteriorated afterwards, possibly due to having thinner than thicker layers of zinc atoms on the sputtered coat (Fig. 2). Physical sputtering of ZSM-5 to achieve enhanced catalysis performance has been reported, *e.g.*, by *Li et al. (2013)* who sputtered ZSM-5 with cobalt nanoparticles, but the application of metal-sputtered zeolite in removal of heavy metals remains largely overlooked. Further investigation of the plasma zinc-sputtering effect is, however, required to further our understanding of its comparative efficiency in terms of copper extraction.

Inferential statistics showed that copper extraction was influenced by treatment ($LR^{X2}_{(5,56)} = 358.01$, $P < 0.0001$), sampling time ($LR^{X2}_{(3,56)} = 1{,}038.73$, $P < 0.0001$), and the interaction between these factors ($LR^{X2}_{(15,56)} = 552.39$, $P < 0.0001$); see Table S5 for a detailed model summary of the levels comprising the main effects.

It is worthy of note that there were no major shifts in pH and conductivity readings over time, but remarkably the application of the zinc-reinforced zeolite −/+ the neodymium magnet led generally to more neutralisation of the acidity of the polluted solution as time passed when compared to the effect of raw zeolite. That was more pronounced when the

*V. faba* bio-trap accompanied the zinc-reinforced zeolite and the magnet, as the pH readings were the closest to neutral (Table S6).

## CONCLUSIONS

The current study produced promising results for copper extraction from a contaminated aquatic system using ZSM-5 zeolite, reinforced with zinc powder, with and without a strong neodymium magnet and/or an aquacultured *V. faba as a* bio-trap. Compared to the raw ZSM-5, the highest and most rapid $Cu2^+$ extraction was by the zinc-reinforced ZSM-5 at 5 min (16% higher than the raw ZSM-5), followed by the bio-trap (14% higher than the raw ZSM-5). By the end of the experiment, the combination of the zinc-reinforced ZSM-5 and the bio-trap led to the most effective $Cu^{2+}$ extraction (78% higher than the raw ZSM-5), followed by the combination of the zinc-augmented ZSM-5, neodymium magnet, and the bio-trap (69% higher than raw ZSM-5). Physically supporting part of the synthetic zeolite surface with zinc-gel-coating led to considerable improvement in the quantity and speed of copper extractability in the drum. This improvement was due to combining sorption *via* the bare porous surface with displacement and deposition on the zinc-coated surface. Furthermore, desirable synergetic effects were achieved when the bio-trap, being the better companion than the neodymium magnet, was simultaneously applied with the zinc-reinforced zeolite, and that surpassed having the magnet as a sole or additional companion. However, the latter combinations were, at any rate, considerably better at extracting copper than having the raw zeolite on its own. Both the reinforced zeolite and the bio-trap showed stability, particularly for the bio-trap against toxic exposure to copper. From this work, the *V. faba* shows promise as a bio-remediating species that is versatile and integral to engineered de-pollution devices. Our timely and novel composite method of having a molecular sieve (zeolite) supported by a partial reactive metal surface with an organic sieve (bio-trap) offers a cost-effective and easy-to-use alternative to existing conventional extraction routes. Moreover, the zinc-sputtered variant of ZSM-5 showed an interesting potential that requires further research for efficacy improvement over time. These findings highlight the utility of ZSM-5 beyond catalysis and bear a promising potential for targeting different heavy metals across different types of polluted waters.

## ACKNOWLEDGEMENTS

We would like to thank Ms. Shahla Khan, Ms. Gemma Chapman, Dr. Desmond Doocey, Dr. Ali Arafeh, the late Dr. Patrick Hill, and the technical staff from the Faculty of Science and Engineering for their support.

### Funding

The authors received no funding for this work.

### Competing Interests

The authors declare that they have no competing interests.

## Author Contributions

- Mouhammad Shadi Khudr conceived and designed the experiments, analyzed the data, performed the computation work, prepared figures and/or tables, authored or reviewed drafts of the article, and approved the final draft.
- Cristian Baleca performed the experiments, analyzed the data, prepared figures and/or tables, authored or reviewed drafts of the article, and approved the final draft.
- Nasser Alqahtani performed the experiments, analyzed the data, prepared figures and/or tables, and approved the final draft.
- Hassan Alhassawi performed the experiments, analyzed the data, prepared figures and/or tables, and approved the final draft.
- Arthur Garforth conceived and designed the experiments, authored or reviewed drafts of the article, supportive materials and resources, validation, and approved the final draft.
- Gordon Tiddy conceived and designed the experiments, authored or reviewed drafts of the article, validation, review, and approved the final draft.
- Abdullatif Alfutimie conceived and designed the experiments, prepared figures and/or tables, authored or reviewed drafts of the article, supervision, project administration, methodology, validation, and approved the final draft.

## Data Availability

Raw data is available at figshare:

Khudr, M.S.; Baleca, Cristian; Alqahtani, Nasser; Alhassawi, Hassan; Garforth, Arthur; Tiddy, Gordon; et al. (2025). Database for the work titled "Zinc-reinforced ZSM-5 subject to a rare-earth magnet and the presence of a legume yields considerable copper extraction". figshare. Dataset. https://doi.org/10.6084/m9.figshare.22568620.v1.

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
