# Peer review of "Zinc-reinforced ZSM-5 subject to a rare-earth magnet and the presence of a legume yields considerable copper extraction"

_PeerJ Materials Science, doi:10.7717/peerj-matsci.37_

## Round 0.1 · original submission · Major Revisions

The manuscript needs a detailed major revision as per suggestions from both the referees. The manuscript must be critically revised.

Reviewer 1 ·

Basic reporting

see comment

Experimental design

see comment

Validity of the findings

see comment

Additional comments

You need some revisions or clarifications regarding the research methods and results you obtained. see comments.
You need some revisions or clarifications regarding your research methods and results. see comments.
this article is good enough to be published with some improvements

Annotated reviews are not available for download in order to protect the identity of reviewers who chose to remain anonymous.

Reviewer 2 ·

Basic reporting

The only two figures presented in the manuscript are far insufficient, with one schematic diagram and one figure showing the removal efficiencies. Please supplement sufficient results of the adsorbent characterizations (XRD, SEM, IR, etc.) and the extraction in the manuscript.

Experimental design

none

Validity of the findings

1.The quality of FTIR spectra in Fig. S9 is not good and the peaks cannot be well recognized.
2.Please add the scale bar in SEM images.
3.The BET analysis results of ZSM-5 in Table S1 are incorrect. The unit of pore size is not cm3/g. Please also supplement the pore volume data.
4.It is strange that the BET surface area increased after treatment, for example, ZnGZ-M-F, ZnGZ+M-F, ZnGZ-M+F and ZnGZ+M+F.
5.Please add an analysis of the potential mechanism of the enhanced extraction of copper ions.

Additional comments

none

---

## Round 0.2 · accepted · Accept

The manuscript is ready for publication.